# Potential Toxicity and Mechanisms of T-2 and HT-2 Individually or in Combination on the Intestinal Barrier Function of Porcine Small Intestinal Epithelial Cells

**DOI:** 10.3390/toxins15120682

**Published:** 2023-12-04

**Authors:** Weihua He, Jianhua Wang, Mengyi Han, Lihua Wang, Ling Li, Jiahui Zhang, Siqi Chen, Jiayi Guo, Xiaohu Zhai, Junhua Yang

**Affiliations:** 1Institute of Pet Science and Technology, Jiangsu Agri-Animal Husbandry Vocational College, Taizhou 225300, China; 2008020102@jsahvc.edu.cn (W.H.); 1997010187@jsahvc.edu.cn (L.W.); 2007020099@jsahvc.edu.cn (L.L.); 202210101@jsahvc.edu.cn (J.Z.); 202210137@jsahvc.edu.cn (S.C.); 202214904@jsahvc.edu.cn (J.G.); 2Institute for Agri-Food Standards and Testing Technology, Shanghai Academy of Agricultural Sciences, Shanghai 201403, China; wangjianhua@saas.sh.cn (J.W.); h2054426758@163.com (M.H.)

**Keywords:** T-2 toxin, HT-2 toxin, porcine intestinal epithelial cells, inflammation, intestinal barrier

## Abstract

Under natural conditions, T-2 toxin can be easily metabolized to HT-2 toxin by deacetylation, and T-2 and HT-2 are usually co-contaminated in grain and feed at a high detected rate. Our previous information indicated that T-2 toxin could injure the function of the intestinal barrier, but the combined toxicity and mechanism of T-2 and HT-2 on the intestinal cells of porcines are still unknown. Therefore, we aimed to explore T-2 and HT-2 individually and combined on cellular viability, cell membrane integrity, the expression of tight junction-related proteins, and the generation of inflammatory factors in porcine intestinal epithelial cells (IPEC-J2). The results showed that T-2 and HT-2, individually or in combination, could induce a decrease in cell viability, an increase in LDH release and IL-1, IL-6, and TNF-α generation, and a decrease in the anti-inflammatory factor IL-10. Based on the analysis of immunofluorescence staining, real-time PCR, and western blotting, the tight junction protein expressions of Claudin-1, Occludin, and ZO-1 were significantly decreased in the T-2 and HT-2 individual or combination treated groups compared with the control. Furthermore, all the parameter changes in the T-2 + HT-2 combination group were much more serious than those in the individual dose groups. These results suggest that T-2 and HT-2, individually and in combination, could induce an intestinal function injury related to an inflammatory response and damage to the intestinal barrier function in porcine intestinal epithelial cells. Additionally, T-2 and HT-2 in combination showed a synergistic toxic effect, which will provide a theoretical basis to assess the risk of T-2 + HT-2 co-contamination in porcine feed.

## 1. Introduction

T-2 toxin, a sesquigerene compound generated by *F. poae* and *F. sporotrichioides*, is characterized by the highest toxicity in type A trichohecene [1], which is found in cereals and unprocessed foods around the world, including Africa, Europe, Southeast Asia, South America, and China [2]. T-2 toxin is difficult to volatilize and is insoluble in water, but it could be resiliently degraded through hydrolysis, hydroxylation, deepoxidation, and conjugation reactions in diverse animal and plant organisms [3]. Generally, the most common metabolic pathway of T-2 is fast deacetylation of the C-4 site, which is then converted into HT-2 toxin [4].

According to the European Union survey of mycotoxins contamination in food, 57% of 11,022 samples tested positive for DON, 16% of 4166 samples tested positive for NIV, 20% of 3490 samples tested positive for T-2, and 14% of 3032 samples tested positive for HT-2 [5,6]. In addition, DON contamination was more commonly observed in wheat and corn, while NIV, T-2 toxin, and HT-2 toxin are all common contaminants in corn, wheat, and oats [6]. In China, T-2 and HT-2 were detected in grain from Sichuan province; the detection rate and average dose of T-2 were 11.64% and 0.565 g/kg, and that of HT-2 were 49.74% and 3.746 μg/kg, and the highest concentrations of T-2 toxin and HT-2 toxins were 3.332 μg/kg and 34.510 μg/kg. Furthermore, the EU is also considering the allowable limits for the T-2 + HT-2 toxin combination [7]. Following the widespread contamination of mycotoxins, the food chain contaminants group of the European Food Safety Authority defined the tolerable daily intake of T-2 toxin and HT-2 toxin at 100 ng/kg [8]. In 2016 and 2017, the mycotoxins contamination in maize was investigated in Croatia, and fumonisin B1 (85.7%), HT-2 (73.8%), deoxynivalenol (65.5%), fumonisin B2 (64.3%), zearalenone (54.8%), T-2 toxin (28.6%), aflatoxin B1 (2.4%), aflatoxin B2, and aflatoxin G1 (1.2%) were discovered in the mixture contamination, and the combined detection rate of T-2 toxin and HT-2 toxin was 26.2% in the total samples. According to the above information, it can be summarized that the levels of T-2 toxin and HT-2 toxin were wide-ranging, and the two toxins often appear in combination.

A large number of studies have shown that the mycotoxins T-2 and HT-2 have extensive toxic effects on humans and animals [9], which could inhibit protein synthesis and cell proliferation and cause toxicity and damage to the skin, immunological system, liver, reproductive system, and digestive system in vitro and in vivo. Because T-2 toxin and HT-2 toxin share the same chemical structure as epoxysesquiterpenes, their toxicity is relatively similar [10]. However, it is not possible to assess the combined toxicity of mycotoxins based on the individual components, as they may produce additive, synergistic, and antagonistic interactions [11]. In earlier studies, T-2 + HT-2 toxins showed synergistic effects at low doses and antagonistic effects at high doses in porcine Leydig cells, but the reversed effect was found in the ternary combination [12]. The gastrointestinal organ is the first target for possible hazardous effects of mycotoxins after ingestion of contaminated food [13]. At present, most studies on the toxic mechanism of mycotoxins on porcine intestinal epithelial cells focus on individual toxicity, but the mechanism of T-2 toxin and HT-2 toxin individually or combined in treatment on the damage of porcine intestinal epithelial cells is still not clear. On the basis of the previous study, cell viability, the permeability of the cell membrane, the expression and distribution of tight junction proteins, and the secretion of inflammatory factors were observed after T-2 and HT-2 individual or combined exposure. The results of this experiment will further enrich the combined toxic mechanism of mycotoxins and provide a new idea for the prevention and control of contamination.

## 2. Results

### 2.1. Effect of T-2 and HT-2 on Cell Viability and Lactate Dehydrogenase (LDH) Activity

As shown in Figure 1A, cell viability was dramatically reduced in response to T-2 toxin and HT-2 toxin individually or in combination groups compared to the controls (*p* < 0.05). Additionally, cell viability in the T-2 + HT-2 combination group was decreased compared with that in the T-2 toxin and HT-2 toxin single-treated groups (*p* < 0.05). 

The LDH released from cells could be quantitatively detected and was considered an important indicator of cell membrane integrity. As shown in Figure 1B, the LDH from IPEC-J2 cells gradually increased when treated with T-2 toxin and HT-2 toxin individually and in combination (*p* > 0.05 or *p* < 0.05), and the LDH in the combination group was obviously higher compared with that in the T-2 toxin and HT-2 toxin individual groups (*p* < 0.05). 

### 2.2. The Expression of Tight Junction Protein by Immunofluorescence Assay

Following the treatment with T-2 toxin and HT-2 toxin, the distribution of tight junction proteins Claudin-1, Occludin, and ZO-1 in IPEC-J2 cells was observed by a laser scanning confocal microscope. As shown in Figure 2, Figure 3 and Figure 4, the red fluorescence intensity was proportional to the expression of the three proteins. In the control group, the expression of the three proteins was high and distributed evenly, but the distribution of Claudin-1, Occludin, and ZO-1 in the combined toxin-treated group decreased, and the expression in the individual T-2 toxin and HT-2 toxin groups decreased significantly. Additionally, T-2 toxin and HT-2 toxin caused some morphological changes in tight junction proteins, which led to the barrier function damage of cell junctions.

### 2.3. Expression of Tight Junction-Related Protein mRNA by Real-Time PCR

The expression levels of *Claudin-1*, *Occludin,* and *ZO-1* genes were investigated by qRT-PCR, as shown in Figure 5. Compared with the control, the levels of the three tight junction protein mRNA were dramatically reduced in all the treated groups (*p* > 0.05 or *p* < 0.05), and the mRNA relative expression of *Claudin-1*, *Occludin,* and *ZO-1* genes in the T-2 + HT-2 combination group was obviously decreased compared with that in the control, T-2 toxin, and HT-2 toxin individual groups (*p* < 0.05). 

### 2.4. Expression of Tight Junction-Related Proteins by Western-Blot

Following treatment with T-2 and HT-2 for 24 h, the expression of Claudin-1, Occludin, and ZO-1 proteins was investigated by western blotting. The protein levels of Occludin and ZO-1 were significantly lower in the T-2 toxin group than those in the control group (*p* < 0.05). The expression levels of the three proteins were dramatically reduced in the HT-2 individual group and the T-2 + HT-2 combination group compared with those in the control and T-2 individual groups (*p* < 0.05). In addition, the expression levels of Claudin-1, Occludin, and ZO-1 in the T-2 + HT-2 combined group were also higher than the HT-2 individual group (*p* < 0.05, Figure 6). 

### 2.5. Effect of T-2 and HT-2 on the Content of Inflammatory Factors

The concentration change in T-2 and HT-2 on inflammatory factors is shown in Figure 7. Compared with the control, the contents of IL-1, IL-6, and TNF-α in IPEC-J2 cells were significantly increased when T-2 toxin and HT-2 toxin were exposed individually or in combination (*p* < 0.05), and obvious differences were found between the different groups (*p* < 0.05). However, the content of IL-10 in the HT-2 toxin and T-2 + HT-2 combination groups was dramatically decreased compared with the control and T-2 toxin groups (*p* < 0.05). Additionally, the level of IL-10 in the combination group was also obviously lower compared with the individual dose groups of T-2 toxin and HT-2 toxin (*p* < 0.05).

## 3. Discussion

Our study results support the conclusion that exposure to T-2 toxin and HT-2 toxin individually or in combination by IPEC-J2 could induce LDH release, decrease tight junction protein expression, and generate pro-inflammatory factors. The toxicological effect of the combination group was more obvious than that of the individual groups, presenting a toxic synergy.

Generally, a basic barrier is composed of the intestinal epithelium and the luminal environment, which could be used to regulate and prevent the intake of harmful substances [14,15,16]. Furthermore, the tight connections (TJ) in the intestine are constituted by claudins and other transmembrane proteins, which are crucial for preserving the integrity of the intestinal mucosa and intestinal health in both humans and animals [17,18]. These proteins are aggregated and stabilized by zonula occluden 1 (ZO-1) and cytoskeletal proteins at the top and Claudin 1 at the bottom of the lateral membrane [19]. Porcine cells are one of the most sensitive cells to T-2 toxin, and our previous studies proved that T-2 toxin could easily cause barrier function damage to IPEC-J2 cells. In this research, T-2 toxin and HT-2 toxin, individually and in combination, damaged the cell membrane integrity and induced the leakage of large amounts of LDH from the cytoplasm to the outside of the cell, which then destroyed the integrity of the intestinal cells. According to reports, T-2 toxin and HT-2 toxin exposed to broiler hepatocytes could induce an increase in LDH level, which was consistent with our results [20]. Similarly, T-2 toxin exposure also led to an increase in LDH levels in TM3 Leydig cells [21]. However, there was no information reported about the LDH level change when treated with T-2 and HT-2 in combination. This study demonstrated that the level of LDH when treated with T-2 + HT-2 was less than in the individual groups, which suggested that the T-2 + HT-2 combination induced a more serious injury to the barrier function in IPEC-J2 and produced a synergistic effect.

According to laser confocal observation, the localization and shape of the tight junction were all changed by treatment with T-2 toxin and HT-2 toxin for 24 h, and changes in the combined group were even more significant. The anatomical basis for preserving the intestinal epithelium and its barrier function was the mechanism of the intestinal mucosal barrier. The main functional element of the intestinal epithelial barrier is TJ proteins, including ZO-1, Claudin-1, and Occludin, which maintain the intestinal barrier’s biological function by sealing the space between adherent epithelial cells [22]. In this study, the mRNA and protein expression levels of ZO-1, Occludin, and Claudin in IPEC-J2 cells were dramatically lower, showing that the T-2 toxin and HT-2 toxin, individually or combined, could injure the barrier function of IPEC-J2 cells. Similarly, the expression levels of ZO-1, Occludin, and Claudin in the intestines of mice and in human epithelial cells were all significantly decreased after T-2 toxin exposure [23]. All the studies indicated that T-2 toxin and HT-2 toxin could damage the intestinal epithelial barrier function, and the toxicological effect of the T-2 toxin and HT-2 toxin combination is much stronger than that of the individual toxins.

Inflammation plays a vitally important role in modulating the barrier function of the intestinal mucosal immune system [24,25]. Additionally, damage to the intestinal barrier function involves an increase in epithelial permeability, translocation of intraluminal allergens and pathogens, non-specific inflammation, and overstimulation of the intestinal-related immune system [23]. Earlier studies found that the IRE1/XBP1 pathway was activated by T-2 toxin exposure, which then disrupted intestinal mucins in 4-week-old BALB/C mice, increased the generation of proinflammatory cytokines, and induced an inflammation response [26]. In the current study, all the results were similar to earlier studies in that the proinflammatory factors IL-1, IL-6, and TNF-α were increased and the anti-inflammatory factor IL-10 was decreased in the IPEC-J2 cells after T-2 and HT-2 individual and combination treatment, which was consistent with the LDH result. However, there was little information about the toxicological effect of T-2 + HT-2 on the expression of inflammatory factors. Other mycotoxins studies have shown that co-exposure to DON and cevalenol could significantly increase the level of inflammatory factors in intestinal explants, with a synergistic effect promoting inflammation [27]. Additionally, both DON and ZEN exposed to intestinal cells can induce synergistic or additive deleterious effects on the expression of inflammatory factors [28,29]. Our experimental results indicate that the influence of inflammation induced by the T-2 toxin and HT-2 toxin combination was significantly greater than that of the individual treatments. Therefore, we were correct in our hypothesis that T-2 toxin and HT-2 toxin, individually or in combination, could increase the inflammation of IPEC-J2 cells and that the combined group has synergistic effects on intestinal barrier damage in pigs. In general, the major metabolic pathway of T-2 toxin in mammals is rapid deacetylation at C-4, which results in HT-2 toxin formation. Therefore, adequate attention should be paid to the combination of T-2 toxin and HT-2 toxin in animals.

## 4. Conclusions

In summary, our data shows that IPEC-J2 cells treated with T-2 toxin and HT-2 toxin individually or in combination had obviously adverse effects on cell viability and caused an increase in LDH release from the cell. This could damage the barrier function of intestinal epithelial cells by disrupting the expression of tight junction protein and inducing the generation of inflammatory factors. Moreover, the damage to the barrier function caused by the T-2 + HT-2 combination group was more serious than that caused by the individual toxin treatments, which suggests a synergistic effect from the T-2 + HT-2 combination. However, their potential impacts on the intestinal health of swine are still worth further study.

## 5. Materials and Methods

### 5.1. Chemicals

The T-2 toxin and HT-2 toxin powders were supplied by Pribolab (Qingdao, Shandong, China). Dulbecco’s Modified Eagle’s Medium (DMEM), penicillin-streptomycin, 0.25% trypsin cell digestive fluid, fetal bovine serum (FBS), dimethyl sulfoxide (DMSO), and HBSS were obtained from Gibco (Grand Island, NY, USA). The CCK-8 kit was supplied by Beyotime Biotechnology (Shanghai, China). Prime Script RT Master Mix, SYBR Green I real-time PCR, and the Luminous kit were purchased from Takara (Dalian, China). The lactate dehydrogenase (LDH) kit was purchased from Applygen Technologies (Beijing, China). Claudin-1, Occludin, and ZO-1 antibodies and FITC-sheep anti-rabbit IgG were purchased from Proteintech (Rosemont, IL, USA). Β-actin was obtained from Servicebio (Wuhan, China).

### 5.2. Cell Culture

The cellular line of IPEC-J2 from Bio-World (Shang, China) was cultured in DMEM containing 10% FBS and 1% penicillin-streptomycin (100 units/mL–100 μg/mL). The incubator condition of the cells was 37 °C in a humidified chamber with 5% CO_2_. T-2 toxin and HT-2 toxin were dissolved in DMSO to prepare a stock solution with a concentration of 1 mg/mL. The final concentration of DMSO was less than 0.1% in all detected processes.

### 5.3. Determination of Cell Viability

According to the CCK-8 kit, the cell viability was determined. IPEC-J2 cells were digested and collected at the logarithmic growth stage. Cell suspension concentration was calculated and then diluted, and then 5 × 10^3^ cells/pores were seeded into a 96-well plate for 24 h. The cells were incubated with T-2 (3.125 nmol/L), HT-2 (6.25 nmol/L), and T-2 (3.125 nmol/L) + HT-2 (6.25 nmol/L) for 24 h, which was based on our previous experiment (unpublished). In the control group, the cells were cultured in 0.1% DMSO in DMEM, and inverted light microscopy was used to analyze the cell development and morphology. Subsequently, the 100 μL complete culture liquor containing different concentrations of T-2 and HT-2 toxins was discarded, and a 10 μL solution from the CCK-8 kit with 100 μL DMEM was added into each well. The cells were dyed and cultured at 37 °C for 2 h. The absorbance was recorded using the microplate reader iMark (Multiskan Sky), and the data were adjusted to reflect the cells that were not treated. Each group underwent six replicates, and a blank control group (no cells) was set up. Cell activity % = (treatment group OD value – blank group OD value)/(control group OD value – blank group OD value).

### 5.4. Determination of Lactate Dehydrogenase (LDH) Activity

IPEC-J2 cells with a density of 1 × 10^5^ cells were inoculated into 96-well plates, and the cells were exposed to different concentrations of T-2 toxin and HT-2 toxin individually or combined for 24 h, and six replicates were performed for each treatment group. Subsequently, the culture medium of 96 well plates in each group was collected, and LDH release activity was detected according to the instructions of the LDH kit. The OD values were observed by a photometer, and the activity units were calculated according to the standard curve values. LDH release rate (%) = enzyme activity unit measured in cellular culture medium/(enzyme activity unit measured in cell lysate + enzyme activity unit measured in cell culture medium) × 100%.

### 5.5. Detection of ZO-1, Occludin, and Claudin-1 by Immunofluorescence Staining

IPEC-J2 cells were inoculated into 12-well plates with a density of 1.5 × 10^5^. During the logarithmic growth period, different levels of T-2 (3.125 nM), HT-2 (6.25 nM), and T-2 + HT-2 (3.125 nM + 6.25 nM) were added to the culture medium for 24 h, and three replicates were performed for each treatment group. Subsequently, a 4% polyoxymethylene solution was immobilized for 10 min, and then 5% goat serum at room temperature was added to seal for 1 h. Finally, the primary antibodies of Claudin-1 (Proteintech13050-1-AP, 1:500), Occludin (Proteintech27260-1-AP, 1:300), and ZO-1 (Proteintech21773-1-AP, 1:500) were added and incubated at 4 °C overnight, respectively. Furthermore, the cells were incubated with biotinylated goat anti-rabbit IgG (1:1000) for 45 min at room temperature. Finally, 4′, 6-diamidino-2-phenylindole (DAPI) (100 ng/mL) and anti-fluorescence quenchant were used to reveal the immunoreaction, and the images were captured and recorded by a fluorescence microscope (Leica Microsystems Inc., Wetzlar, Germany).

### 5.6. RNA Extraction and Real-Time Quantitative PCR

IPEC-J2 cells were inoculated into 6-well plates with a density of 1 × 10^6^ cells in each well. After 24 h of incubation, the cells were collected from each group of T-2 toxin and HT-2 toxin, individually or in combination, and three replicates were performed for each treatment group. The mRNA expression levels for tight junction-associated genes were determined by qPCR. TRIZOL was used to extract the total RNA, and then a Prime Script RT Master Mix kit was used to perform the cDNAs. According to the nucleotide sequences of *Claudin-1*, *Occludin,* and *ZO-1* to synthesize primers (shown in Table 1), *β-actin* was selected as the reference gene. A real-time qPCR reaction was performed according to the manufacturer’s instructions for the SYBR Green I kit, and the 2^−ΔΔCT^ method was used to analyze the relative expression of each gene [30].

### 5.7. Detection of Tight Junction-Associated Proteins

After washing the cell, RIPA protein lysate was used to extract the total protein, and the BCA kit was performed to determine the concentration of protein. Subsequently, 15 μg of protein was separated by 12.5% sodium dodecyl sulfate polyacrylamide gel electrophoresis (SDS-PAGE). The separated proteins were transferred onto a nitrocellulose (NC) membrane, and then 5% skim milk powder was used to block the membrane. After washing with TBS, membranes were probed with anti-Claudin-1 (Proteintech13050-1-AP, 1:750), anti-Occludin (Proteintech27260-1-AP, 1:750), anti-ZO-1 (Proteintech21773-1-AP, 1:750), and anti-β-actin (ServicebioGB12001, 1:3000) antibodies at 4 °C overnight. After that, the blots were incubated with horseradish peroxidase (HRP)-conjugated secondary antibody (Thermo Scientific, Waltham, MA, USA, 1:10,000) at room temperature for 1 h. Finally, the signals were captured, and the intensities of proteins on the bands were quantified using ImageJ Software (Java 1.8.0_112, National Institute of Health, Bethesda, MD, USA), and the expression of the target protein was analyzed using β-actin as an internal reference.

### 5.8. Effect of T-2 and HT-2 on Inflammation Factors

The levels of IL-1, IL-6, IL-10, and TNF-α were detected according to the ELISA kit from MLBIO (Shanghai Enzyme-linked Biotechnology Co., Ltd., Shanghai, China). IPEC-J2 cells were seeded at a density of 1 × 10^5^ cells per hole on a 24-well plate. After incubation with T-2 (3.125 nmol/L), HT-2 (6.25 nmol/L), and T-2 (3.125 nmol/L) + HT-2 (6.25 nmol/L) for 24 h, the supernatant was centrifuged and collected to remove impurities and cell fragments. Based on the competitive ELISA, a standard curve was set, and the samples meeting calibration standards were assayed according to the manufacturer’s instructions. The levels of inflammation factors were calculated by comparing the OD values of the samples to their respective standard curves at 450 nm within 30 min. Six replicates were performed for each treatment group.

### 5.9. Statistical Analysis

SPSS 17.0 (SPSS Inc., Chicago, IL, USA) was used for all analyses. Statistical analyses were performed by one-way ANOVA and LSD’s post-hoc test. The data were presented as mean ± SEM, and *p* < 0.05 was considered statistically significant.

## Figures and Tables

**Figure 1 toxins-15-00682-f001:**
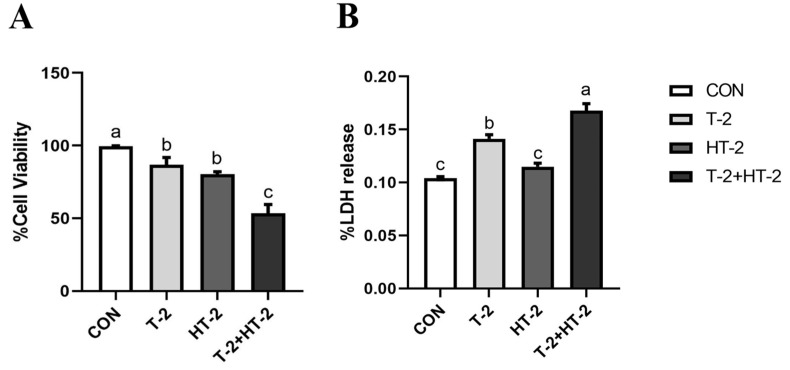
Changes in cell viability and LDH release of IPEC-J2 cells after exposure to T-2 (3.125 nmol/L), HT-2 (6.25 nmol/L), and T-2 (3.125 nmol/L) + HT-2 (6.25 nmol/L) for 24 h. (**A**) Cell viability; (**B**) lactate dehydrogenase activity. All values are given as the mean ± SD of four independent experiments. Values with different superscript letters are significantly different (*p* < 0.05).

**Figure 2 toxins-15-00682-f002:**
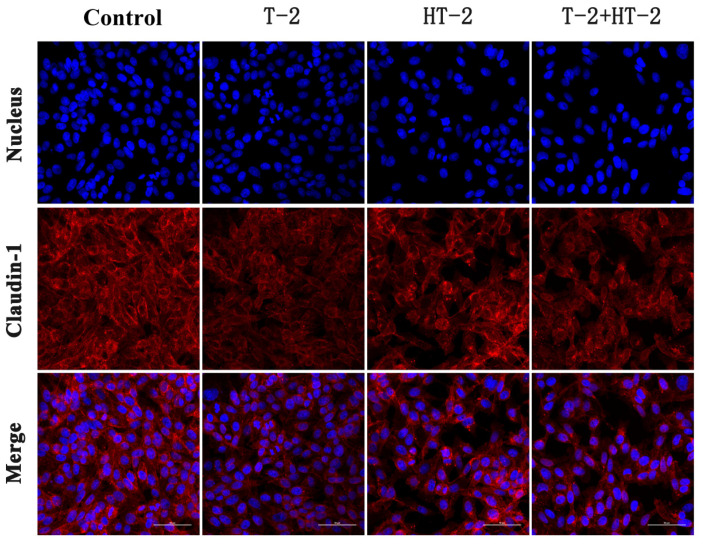
Changes in expression and distribution of Claudin-1 protein in IPEC-J2 cells after exposure to T-2 (3.125 nmol/L), HT-2 (6.25 nmol/L), and T-2 (3.125 nmol/L) + HT-2 (6.25 nmol/L) for 24 h (400×).

**Figure 3 toxins-15-00682-f003:**
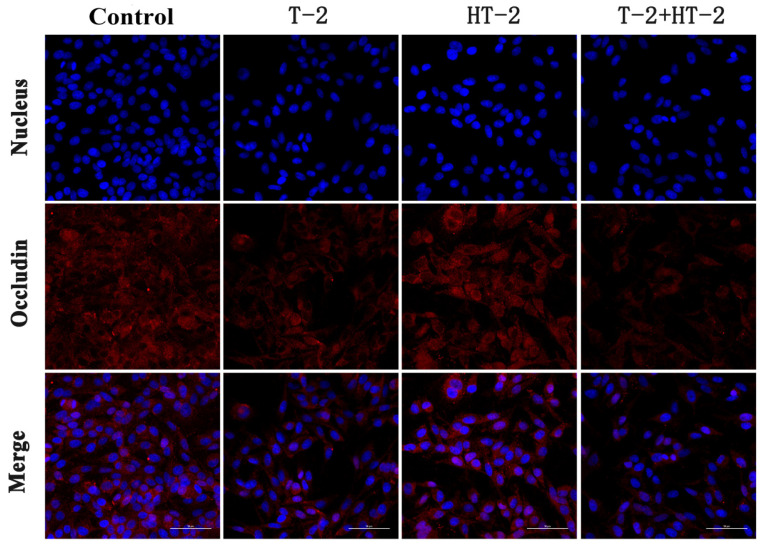
Changes in expression and distribution of Occludin protein in IPEC-J2 cells after exposure to T-2 (3.125 nmol/L), HT-2 (6.25 nmol/L), and T-2 (3.125 nmol/L) + HT-2 (6.25 nmol/L) for 24 h (400×).

**Figure 4 toxins-15-00682-f004:**
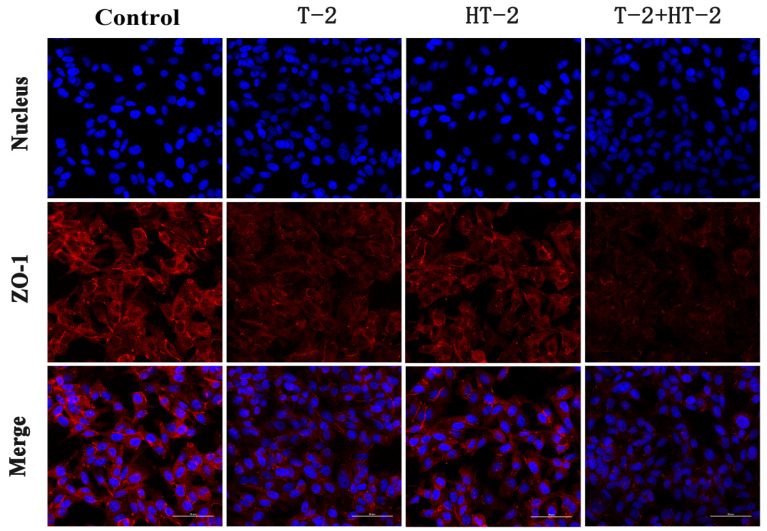
Changes in expression and distribution of ZO-1 protein in IPEC-J2 cells after exposure to T-2 (3.125 nmol/L), HT-2 (6.25 nmol/L), and T-2 (3.125 nmol/L) + HT-2 (6.25 nmol/L) for 24 h (400×).

**Figure 5 toxins-15-00682-f005:**
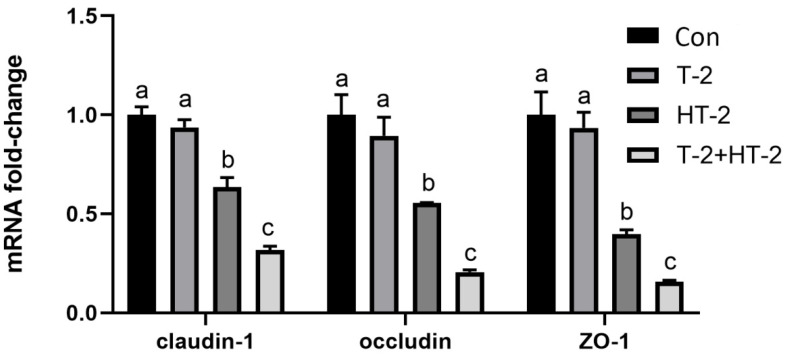
Changes in expression level of tight junction-related genes in IPEC-J2 cells after exposure to T-2 (3.125 nmol/L), HT-2 (6.25 nmol/L), and T-2 (3.125 nmol/L) + HT-2 (6.25 nmol/L) for 24 h. The control value was set at 1.0. All values are given as the mean ± SD of four independent experiments. Values with different superscript letters are significantly different (*p* < 0.05).

**Figure 6 toxins-15-00682-f006:**
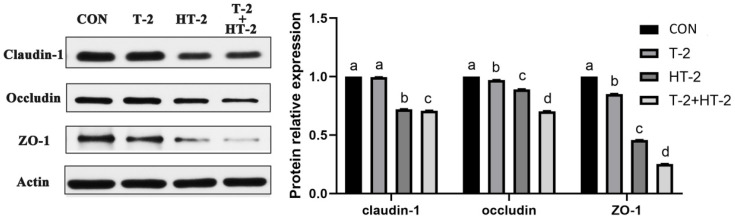
Changes in relative expression of tight junction-related proteins in IPEC-J2 cells after exposure to T-2 (3.125 nmol/L), HT-2 (6.25 nmol/L), and T-2 (3.125 nmol/L) + HT-2 (6.25 nmol/L) for 24 h. The control value was set at 1.0. All values are given as the mean ± SD of four independent experiments. Values with different superscript letters are significantly different (*p* < 0.05).

**Figure 7 toxins-15-00682-f007:**
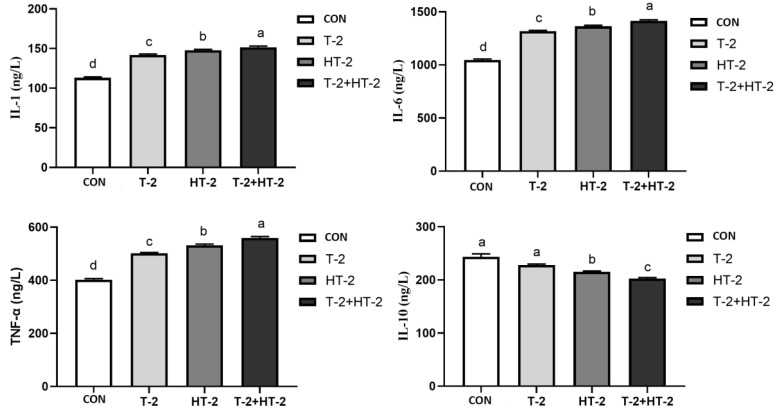
Changes in content of IL-1, IL-6, TNF-a, and IL-10 in IPEC-J2 cells after exposure to T-2 (3.125 nmol/L), HT-2 (6.25 nmol/L), and T-2 (3.125 nmol/L) + HT-2 (6.25 nmol/L) for 24 h. All values are given as the mean ± SD of four independent experiments. Values with different superscript letters are significantly different (*p* < 0.05).

**Table 1 toxins-15-00682-t001:** Parameters of primer for tight junction-related genes and β-actin genes.

Gene	Accession Number	Sequences (5′→3′)	Product
*Claudin-1*	NC_010455.5	F: GGCAGATCCAGTGCAAAGTC R: CCCAGCAGGATGCCAATTAC	94 bp
*Occludin*	NC_010458.4	F: CATTATGCACCCAGCAACGA R: GCACATCACGATAACGAGCA	168 bp
*ZO-1*	NC_010443.5	F: GGGCTCTTGGCTTGCTATTC R: AAGGCCTCGGAATCTCCAAA	160 bp
*β-actin*	424396	F: CTGGACTTCGAGCAGGAGATGG R: TTCGTGGATGCCGCAGGATTC	168 bp

Note: qPCR procedures were used as follows: pre-denaturation (95 °C 30 s), PCR reaction (95 °C 5 s, 60 °C 34 s, 40 cycles), 95 °C 15 s, and 60 °C 45 s. Each sample gene and *β-actin* were amplified under the same conditions and repeated 3 times.

## Data Availability

Data are contained within the article.

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
