# Peer review of "Potential Toxicity and Mechanisms of T-2 and HT-2 Individually or in Combination on the Intestinal Barrier Function of Porcine Small Intestinal Epithelial Cells"

_toxins, 2023, doi:10.3390/toxins15120682_

Round 1

Reviewer 1 Report

Comments and Suggestions for Authors

Comments/suggestions:

Line 34: 1 : Literature item is missing from the citation before item 2 - please check citations throughout the manuscript.
Line 44: Please correct "T-2" instead of "T2".
Line 47: Please correct "th1e" to "the".
Line 166: claudin-1 and zonula or claudin-1 and ZO-1? Please correct this.

No reference to authors' contribution and funding, please complete relevant information according to journal requirements.

Reference to full review, no strict reference according to journal requirements

The reviewed manuscript is interesting and demonstrates the effect of T-2 and HT-2 toxin exposure on damage to intestinal barrier function, and consequently on the inflammatory response and decreased protein expression, as well as confirms the synergistic effect of the toxic combination of T-2 and HT-2 toxins. The documentation of the work, discussion and conclusions are not objectionable. The remaining minor shortcomings will certainly be corrected by the Authors during the reference to the comments and suggestions.

Comments on the Quality of English Language

Minor eding of English language required

Author Response

Dear Editors and Reviewers:

Thanks very much for your letter and for the reviewers’ comments concerning our manuscript entitled “Potential toxicity and mechanisms of T-2 and HT-2 individual or combination on the intestinal barrier function of porcine small intestinal epithelial cells” (ID: toxins-2704479). Those comments are all valuable and very helpful for revising and improving our paper, as well as the important guiding significance to our research. We have studied comments carefully and have made correction which we hope meet with approval. The main corrections in the paper and the responds to the reviewer's comments are as following:

Responds to the editor's comments:

Question 1: In addition, generally we request there are more than 4000 words in the main text and the references could be more than 30. Please expend them during the revision.

Response: Thank you so much for editor’s comments, and we have made the corresponding supplements in all manuscript.

Question 2: The ithenticate rate of all reference is 47%. The ithenticate rate of single reference is 12% (report attached). Generally, we hope the ithenticate rate of all reference is less than 30%, the ithenticate rate of single reference is less than 5%. So please revise them during revision)

Response: Special thanks for reviewer’s comments. We have made all the correction according to the suggestion. And we hope the revised manuscript could be acceptable for you.

Responds to the reviewer's comments:

Reviewer #1:

The reviewed manuscript is interesting and demonstrates the effect of T-2 and HT-2 toxin exposure on damage to intestinal barrier function, and consequently on the inflammatory response and decreased protein expression, as well as confirms the synergistic effect of the toxic combination of T-2 and HT-2 toxins. The documentation of the work, discussion and conclusions are not objectionable. The remaining minor shortcomings will certainly be corrected by the Authors during the reference to the comments and suggestions.

Question 1: Line 34: 1. Literature item is missing from the citation before item. Response: Thanks very much for the reviewer’s good suggestions. We have made supplication of the citation according to the Reviewer’s comments,

Question 2: “ – ”please check citations throughout the manuscript.

Response: Special thanks for reviewer’s comments. We have made all correction throughout the manuscript according to the suggestion. And we hope the revised manuscript could be acceptable for you.

Question 3: Line 44: Please correct "T-2" instead of "T2".

Response: We sincerely appreciate the valuable comments. And we have made correction according to the reviewer’s suggestion. Thanks again for reviewer’s nice comments.

Question 4: Line 47: Please correct "th1e" to "the".

Response: Thanks very much for the reviewer’s good comments. And we have made the correction, please forgive our carelessness.

Question 5: Line 166: claudin-1 and zonula or claudin-1 and ZO-1? Please correct this.

Response: Special thanks for reviewer’s comments. And we have made correction according to the reviewer’s suggestion. Thanks again for reviewer’s nice comments.

Question 6: No reference to authors' contribution and funding, please complete relevant information according to journal requirements.

Response: Thanks very much for the reviewer’s good suggestions. We have made supplication according to the Reviewer’s comments.

Question 7: Reference to full review, no strict reference according to journal requirements

Response: Thanks very much for the reviewer’s good suggestions. We have made correction and rewritten this section according to the journal requirements

We tried our best to improve the manuscript and made some changes in the manuscript. These changes will not influence the content and framework of the paper.

We appreciate for Editors/Reviewers' warm work earnestly, and hope that the correction will meet with approval.

Once again, thank you very much for your comments and suggestions.

If there appears any question, please do not hesitate to contact me. We look forward to hearing from you.

Warm regards

Reviewer 2 Report

Comments and Suggestions for Authors

I will have few comments to be addressed:

1.     Some of the keywords are repeated. Select other terms to increase discoverability

2.     Figures 1 and 2 are both about cytotoxicity and would be better combined into one Figure.

3.     Lines 71-81 should be deleted, the contents belong to the part of “Materials and Methods”, not” Introduction”.

4.     The fluorescence of Figures 3, 4, and 5 was not quantified relatively, which is not conducive to understanding the results, please add the relative quantification of the fluorescence.

5.     Line 122The names of genes should be italic. Please check it throughout the manuscript.

6.     Lines 271-272: The Source and Specificity of primary antibodies should be provided in the materials and methods.

7.     Each Figure is independent, please add information on the concentration of the toxin used in each figure as well as the N-value throughout the manuscript.

8.     Lines 268-269:Why the concentration of the T-2 toxin in Immunofluorescence Staining is different from other experiments, for example, CCK8 and LDH

9.     Were the experiments done in duplicate or triplicate? Please add it in the part of Materials and Methods

10.  In the discussion section, metabolisms of T-2 and HT-2 in animals should be explained in more detail.

Comments on the Quality of English Language

Minor editing of English language required

Author Response

Dear Editors and Reviewers:

Thanks very much for your letter and for the reviewers’ comments concerning our manuscript entitled “Potential toxicity and mechanisms of T-2 and HT-2 individual or combination on the intestinal barrier function of porcine small intestinal epithelial cells” (ID: toxins-2704479). Those comments are all valuable and very helpful for revising and improving our paper, as well as the important guiding significance to our research. We have studied comments carefully and have made correction which we hope meet with approval. The main corrections in the paper and the responds to the reviewer's comments are as following:

Responds to the reviewer's comments:

Reviewer #2:

Question 1: Some of the keywords are repeated. Select other terms to increase discoverability

Response: Special thanks for reviewer’s comments. And we have rewritten the keywords according to the reviewer’s suggestion. Thanks again for reviewer’s nice comments.

Question 2: Figures 1 and 2 are both about cytotoxicity and would be better combined into one Figure.

Response: Thanks very much for the reviewer’s good suggestions. We have combined the two figures according to the Reviewer’s comments.

Question 3: Lines 71-81 should be deleted, the contents belong to the part of “Materials and Methods”, not” Introduction”.

Response: We sincerely appreciate the valuable comments. And we have deleted the Lines 71-81 according to the reviewer’s suggestion. Thanks again for reviewer’s nice comments.

Question 4: The fluorescence of Figures 3, 4, and 5 was not quantified relatively, which is not conducive to understanding the results, please add the relative quantification of the fluorescence.

Response: Thanks very much for the reviewer’s good suggestions. At the beginning, we thought that the results from relative quantification of the fluorescence were used for position observation and qualitative analysis, and the RT-qPCR and Western blotting were utilized to quantification. Combining the results of these three parts, we can obtain an accurate expression of related proteins.

Question 5: Line 122:The names of genes should be italic. Please check it throughout the manuscript.

Response: We sincerely appreciate the valuable comments. And we have made the correction of genes format throughout the manuscript.

Question 6: Lines 271-272: The Source and Specificity of primary antibodies should be provided in the materials and methods.

Response: Thanks very much for the reviewer’s good suggestions. We have made the supplement of information of primary antibodies in the materials and methods. Thanks again for reviewer’s nice comments.

Question 7: Each Figure is independent, please add information on the concentration of the toxin used in each figure as well as the N-value throughout the manuscript.

Response: Special thanks for reviewer’s suggestions. We have made the supplement of information on the concentration of the toxin in each figure. Thanks again for reviewer’s nice comments.

Question 8: Lines 268-269:Why the concentration of the T-2 toxin in Immunofluorescence Staining is different from other experiments, for example, CCK8 and LDH?

Response: Thanks very much for the reviewer’s good suggestions. The concentration of T-2 toxin in the whole manuscript was the same, and we have made the correction, please forgive our carelessness. Thanks again for reviewer’s nice reminders

Question 9: Were the experiments done in duplicate or triplicate? Please add it in the part of Materials and Methods

Response: We sincerely appreciate the valuable comments. And we have made the supplement in the part of Materials and Methods. Thanks again for reviewer’s nice comments.

Question 10: In the discussion section, metabolisms of T-2 and HT-2 in animals should be explained in more detail.

Response: Thanks very much for the reviewer’s good suggestions. We have made supplication according to the Reviewer’s comments, which will clear explain the metabolisms of T-2 and HT-2 in animals.

Question 11: Comments on the Quality of English Language, Minor editing of English language required

Response: Special thanks for reviewer’s comments. We have sought the English language editing assistance and made all correction according to the suggestion.

We tried our best to improve the manuscript and made some changes in the manuscript. These changes will not influence the content and framework of the paper.

We appreciate for Editors/Reviewers' warm work earnestly, and hope that the correction will meet with approval.

Once again, thank you very much for your comments and suggestions.

If there appears any question, please do not hesitate to contact me. We look forward to hearing from you.

Warm regards

Round 2

Reviewer 2 Report

Comments and Suggestions for Authors

All comments from me had been responded one by one and the manuscript had been polished. So I suggested to accept it in current form.